# The Bateman-Type Soft Tissue Reconstruction around Proximal or Total Humeral Megaprostheses in Patients with Primary Malignant Bone Tumors—Functional Outcome and Endoprosthetic Complications

**DOI:** 10.3390/cancers13163971

**Published:** 2021-08-05

**Authors:** Helmut Ahrens, Christoph Theil, Georg Gosheger, Robert Rödl, Niklas Deventer, Carolin Rickert, Thomas Ackmann, Jan Schwarze, Sebastian Klingebiel, Kristian Nikolaus Schneider

**Affiliations:** Department of Orthopaedics and Tumor Orthopaedics, Albert-Schweitzer Campus 1, University Hospital of Münster, 48149 Münster, Germany; helmut.ahrens@ukmuenster.de (H.A.); christoph.theil@ukmuenster.de (C.T.); georg.gosheger@ukmuenster.de (G.G.); robert.roedl@ukmuenster.de (R.R.); niklas.deventer@ukmuenster.de (N.D.); carolin.rickert@ukmuenster.de (C.R.); thomas.ackmann@ukmuenster.de (T.A.); jan.schwarze@ukmuenster.de (J.S.); sebastian.klingebiel@ukmuenster.de (S.K.)

**Keywords:** primary malignant bone tumors, limb-salvage, megaprosthesis, soft tissue, functional outcome, endoprosthetic complications

## Abstract

**Simple Summary:**

The soft tissue reconstruction around proximal or total humeral megaprostheses following limb-sparing resection of a primary malignant bone tumor can be a challenge. A surgical technique to overcome these challenges is the Bateman-type soft tissue reconstruction, which is performed as a lateral acromion and trapezoid transfer aiming to improve soft tissue coverage and postoperative function. However, the functional outcome and the endoprosthetic complications of this procedure have hardly been evaluated. Our study shows that the Bateman-type reconstruction is a feasible treatment option, but the postoperative functional outcome is overall limited although good to excellent functional results can be achieved in individual patients. The risk for revision surgery is high within the first year, but remains low thereafter.

**Abstract:**

We aimed to evaluate the functional outcome and endoprosthetic complications following the Bateman-type soft tissue reconstruction around proximal or total humeral replacements in patients undergoing resection of a primary malignant bone tumor. Between September 2001 and December 2018, a total of 102 patients underwent resection of a primary malignant bone tumor and subsequent reconstruction with a modular humeral megaprosthesis in our department. Fifteen (15%) of these patients underwent a Bateman-type soft tissue reconstruction and were included in this retrospective study. The median Musculoskeletal Tumor Society (MSTS) score was 21, the median Toronto Extremity Salvage Score (TESS) was 70, and the median American Shoulder and Elbow Surgeons (ASES) score was 72. Fifty-three percent (8/15) of all patients required a revision surgery after a median time of 6 months. There were 2 soft tissue failures, 3 infections and 3 tumor recurrences. The revision-free implant survivorship amounted to 53% (95% confidence interval (CI) 28–81) after 1 year and 47% (95% CI 22–73) at last follow-up. The Bateman-type reconstruction is a feasible option for soft tissue reconstruction but functional outcome is overall limited and the risk for revision surgery within the first postoperative year is high.

## 1. Introduction

The humerus is a common site for primary malignant bone tumors [1]. While there are several multimodal treatment protocols that are adapted depending on tumor histology, tumor resection with a wide margin is considered the gold standard surgical treatment [2]. In the majority of cases, a limb-sparing resection is possible, but surgeons are subsequently challenged with the reconstruction of the segmental bone defect and the adjacent shoulder joint [3]. Over the last decades, endoprosthetic replacement using a modular megaprosthesis has evolved as a favorable and widely used approach for reconstruction [4,5,6]. However, while long-term implant survivorship of proximal or total humeral replacements can be considered good, soft tissue failures such as dislocation, migration and problematic implant coverage are the main cause of revision surgery [5,7,8]. Furthermore, functional outcome in these patients can be limited [9,10,11,12].

Surgeons have attempted several surgical techniques using muscle transfers and en-bloc flaps to address these challenges [11,13,14,15,16,17] (Table 1). One surgical procedure is the Bateman procedure, which was originally described by Bateman J.E. in 1955 for patients with an irreparable paralysis of the deltoid muscle due to axillary nerve damage [18]. In these patients, Bateman aimed to restore shoulder abduction by utilizing the trapezius muscle with its muscle attachments at the acromion and the lateral part of the scapular spine, transferring and fixing it as far down the humeral shaft as possible, maximizing leverage [18].

However, despite the need for improved techniques in soft tissue reconstruction around proximal and total humeral megaprostheses, this procedure has only been investigated by a single case report [17].

Thus, the aim of our study was to evaluate the functional outcome and endoprosthetic complications of a Bateman-type reconstruction performed as a lateral acromion and trapezoid transfer in patients with primary malignant bone tumor who underwent resection and megaprosthetic proximal or total humerus replacement.

## 2. Materials and Methods

Between September 2001 and December 2018, a total of 580 patients underwent resection of a primary malignant tumor of the long bones and subsequent reconstruction with a modular megaprosthesis in our department. In 102 of these patients (18%), location of the tumor was the proximal humerus. 15 of these patients (15%) underwent the Bateman-type reconstruction and were included in this retrospective study (Figure 1, Table 2).

### 2.1. Data Collection

Patients’ demographics, tumour characteristics, surgical and oncological treatment as well as endoprosthetic failure and its subsequent treatment were retrospectively retrieved from patients’ electronic medical records. Functional outcome was determined in surviving patients at last follow-up and included evaluation of the shoulder range of motion (ROM) and determination of 3 standardized scoring systems: the Musculoskeletal Tumor Society (MSTS) score, the Toronto Extremity Salvage Score (TESS) and the American Shoulder and Elbow Surgeons Score (ASES) [20,21,22].

### 2.2. Study Population

Fifteen patients (9 male) with a median BMI of 25 (Interquartile range (IQR) 22–27) and a median age at the time of surgery of 19 (IQR 15–34) years were available for analysis after a median follow-up of 49 (IQR 25–100) months. Patients who died of their disease or underwent revision surgery were included irrespective of the follow-up period; for all other patients the minimum follow-up was 24 months. Seven patients died of their disease after a median time of 25 (IQR 20–49) months. In the 8 surviving patients, the median follow-up amounted to 80 (IQR 47–113) months. Functional outcome data were available for 6 of these patients as 1 patient lived overseas and 1 patient was not contactable. The overall patient survival probability was 80% (95% CI 60–100) after 2 years and 50% (95% CI 24–76) after 5 years. Demographic, oncological and surgical details are presented in Table 2.

### 2.3. Surgical Technique

All patients undergoing the Bateman-type reconstruction (Figure 2) were treated under general anaesthesia and in the lateral decubitus position. To ensure a sufficient anatomic exposure, an anterolateral approach with resection of the biopsy channel was established. Depending on tumour extension on preoperative imaging, neurovascular structures, muscle tissue and humeral bone were preserved, respecting sufficient margins. The respective tendons of the deltoid muscle, triceps, biceps and rotator cuff were tagged. Following glenohumeral dislocation and removal of the tumor, a reamer was used to prepare the remaining humeral shaft in patients undergoing proximal humeral replacement or to prepare the proximal ulna in patients undergoing total humeral replacement for subsequent press-fit implantation of the modular tumor endoprothesis (Figure 3a). In patients with reduced bone quality and where a press-fit fixation was not stable, a cemented humeral/ulnar stem fixation was used. Depending on the resection of the rotator cuff, deltoid muscle and preservation of the axillary nerve, a reverse or anatomic shoulder reconstruction was performed [23,24]. Additionally, an attachment tube (Trevira tube, Implantcast GmbH, Buxtehude; Germany) was used for soft tissue reconstruction and was fixed to the glenoid with non-absorbable No. 2 FiberWire^®^ (Arthrex, Naples, FL, USA) [25,26] (Figure 3b). The modular megaprosthesis was then assembled and shuttled through the attachment tube. To achieve glenohumeral stability, several FiberWire^®^ sutures were used to fixate the attachment tube to the implant (Figure 3c). If there was insufficient coverage of the implant and the attachment tube after mobilization and refixation of the remaining shoulder and arm muscles following tumor resection, a Bateman-type transfer was performed as the reconstructive approach of choice and no other techniques for soft tissue coverage around humeral megaprostheses were used during that time. While in previous cases a combination of the Bateman-type reconstruction and an additional latissimus dorsi transfer was performed, non-satisfactory functional results led to the sole performance of the Bateman-type reconstruction in the present cohort [17]. For the Bateman-type reconstruction, the trapezius muscle with its acromial bony attachment was identified, freed from deltoid muscle fibers and then mobilized and attached to the attachment tube at the lateral proximal humerus using several FiberWire^®^ sutures (Figure 3c,d). Afterwards, the previously tagged tendons and remaining soft tissue were attached to the attachment tube [25,26]. Postoperatively, the arm was immobilized in an abduction orthesis (DONJOY^®^ Ultrasling^®^ III, Ormed GmbH, Freiburg, Germany) for 6 weeks. Active mobilization of the hand and elbow were encouraged immediately on the second postoperative day, whilst passive mobilization of the shoulder started 4 weeks postoperatively with increasing active and assistive mobilization following 6 weeks postoperatively.

### 2.4. Classification of Endoprosthetic Failure and Assessment of Functional Outcome

Endoprosthetic failure was classified according to Henderson et al., as soft tissue failure (Type 1), aseptic loosening (Type 2), structural failure (Type 3), infection (Type 4) and tumour progression (Type 5) [7]. Functional outcome was determined using 3 standardized scoring systems: The physician-reported MSTS score, which is specifically developed for functional evaluation following limb-salvage surgery in sarcoma patients and consists of 6 categories in the upper extremity version, each rated from 0 (worst) to 5 (best): pain, function, emotional acceptance, hand positioning, dexterity and lifting ability [20]. The patient-reported TESS upper extremity questionnaire, which is also specifically developed for functional evaluation of sarcoma patients undergoing limb-salvage surgery, was also used [21]. For the TESS, patients rate their functional difficulties experienced by performing a total of 29 daily tasks including dressing, self-care, household chores, mobility, leisure and work activities on a scale from 1 (impossible to do) to 5 (not at all difficult) [21,27]. The patient- and physician-reported ASES score, which is a more general score developed to assess functional outcome after shoulder and elbow surgery is equally weighted for pain (50%) and function (50%) [22]. While the pain is assessed using a VAS, function is determined by 10 questions regarding daily and sporting activities, each scored from 0 (unable to do) to 3 (not difficult) [22].

### 2.5. Statistical Analysis

Statistical analysis was performed using SPSS 25.0 (IBM Corp., Armonk, NY, USA). The duration of follow-up and the time to event were calculated from the date of surgery to the date of event or the last documented contact with the patient as of May 2021. Implant and patient survival probabilities with 95% confidence intervals (CI) were calculated using the Kaplan–Meier method and compared with the log-rank test. Depending on data distribution, as determined by the Kolmogorov–Smirnov test, non-parametric analyses were performed with the Mann–Whitney-U-test and parametric comparisons were made with the student’s *t*-test. All *p*-values are two-sided and a *p*-value < 0.05 was considered to be statistically significant.

### 2.6. Ethics and Funding

The study was approved by the local ethics committee (Ethik-Kommission der Ärztekammer Westfalen-Lippe, 2020-898-f-S) and performed in accordance with the Declaration of Helsinki. We acknowledge support from the Open Access Fund of the University of Münster.

## 3. Results

### 3.1. Functional Outcome

The median MSTS score amounted to 21 (IQR 17–27), the median TESS to 70 (IQR 59–91) and the median ASES to 72 (IQR 64–85). The median shoulder anteversion was 20° (IQR 10–38°) with a 20° median retroversion (IQR 10–25°), 20° median abduction (IQR 10–30°), 20° median adduction (IQR 10–30), 40° median internal rotation (IQR 25–40°) and 20° median external rotation (IQR 10–35). The best functional outcome scores (MSTS 28, ASES 95, TESS 97) were observed at the 10-year-follow-up examination in a then 27-year-old male patient who underwent extra-articular resection of an osteosarcoma and subsequent proximal humeral replacement (Appendix A).

No differences in functional outcome scores were found for the 3 patients undergoing an intra- or extra-articular tumor resection, respectively or the 3 patients who underwent an anatomic or reverse shoulder reconstruction, respectively (Table 3). Furthermore, no differences in shoulder ROM were found among patients with intra- and extra-articular tumor resection, while for patients undergoing reverse shoulder reconstruction all ROM except internal rotation were better compared to patients undergoing anatomic shoulder reconstruction (Table 4).

### 3.2. Endoprosthetic Complications

Fifty-three percent (8/15) of all patients required revision for a complication after a median time of 6 months (IQR 3–10). There were 2 soft tissue failures (Henderson Type 1; 1 dislocation, 1 wound dehiscence) that were treated with a modular component exchange and a soft tissue revision, respectively. There were 3 infections (Henderson Type 4) that were treated with debridement, exchange of mobile components and antibiotics in the first patient, removal of the prosthesis and insertion of a spacer in the second patient and the third patient underwent amputation due to poor soft tissues. Furthermore, there were 3 cases of tumor recurrence (Henderson Type 5) after a median time of 7 months. Two of these patients underwent amputation and a limb-sparing surgery was performed in the remaining patient due to a small soft tissue lesion. However, the latter patient had a re-recurrence 52 months after the initial surgery and 45 months after the first recurrence, which was treated by an amputation. Overall, the revision-free implant survivorship amounted to 53% (95% CI 28–81) after 1 year and to 47% (95% CI 22–73) at last follow-up. The limb survival probability was 93% (95% CI 81–100) at 1 year and 72% (95% CI 32–100) at last follow-up.

## 4. Discussion

The most important findings of our study are: (1) Functional outcome following the Bateman-type procedure in musculoskeletal oncology is limited but good to excellent results as determined by MSTS, TESS and ASES score can be achieved in individual patients. (2) The revision-free implant survivorship after 1 year is diminished with nearly every second patient requiring revision surgery during the first 12 months. (3) Limb survivorship at 1 year and at final follow-up is overall acceptable.

Several previous studies have evaluated the functional outcome of Bateman-type procedures in non-oncological cohorts [28,29]. Agrawal et al. performed a modified Bateman procedure by mobilizing the trapezius only with its clavicular attachment in 32 patients with a brachial plexus injury and reported an improvement of the mean shoulder anteversion from 6° (range, 0–15°) to 55° (range, 40–90°) and an improved mean shoulder abduction from 8° (range, 0–30°) to 85° (range, 45–140°) [28].

Functional results in our cohort remain overall lower than those previously reported. But even Agrawal et al. pointed out that “functional outcome following transfer of the trapezius muscle varies considerably” [28]. While the sole aim of performing the Bateman procedure in non-oncological patients is to improve function, the aim in oncological patient cohorts is also soft tissue coverage, resulting in a trade-off of the optimal anatomic leverage, which subsequently results in a poorer functional outcome and a higher complication rate. Furthermore, all of the patients in our cohort underwent glenohumeral joint replacement making it difficult to compare function and complications between oncological and non-oncological patient cohorts.

One different but similar surgical approach being frequently performed in musculoskeletal oncology is the Tikhoff–Linberg procedure [13,14]. The classic Tikhoff–Linberg procedure is defined as an en bloc removal of the scapula, the lateral clavicle and the proximal humerus and the respective connecting muscles [13]. Later, various modifications to this procedure were reported, including endoprosthetic replacement of the proximal humerus and resection of only the glenoid while saving the remaining scapula [13,14]. Capanna et al. performed 12 classic and 12 modified Tikhoff–Linberg procedures in 24 patients with malignancies of the proximal humerus [14]. While not reporting average values for the shoulder ROM, Capanna et al. have assessed function using a modified Enneking functional evaluation system rating 1 patient good, 6 fair and 5 poor from the cohort of the classic Tikhoff–Linberg procedure and 4 good, 7 fair and 1 poor in the cohort undergoing the modified Tikhoff–Linberg procedure [14].

Voggenreiter et al. performed the modified Tikhoff–Linberg procedure in 19 patients with primary malignant bone tumors, malignant soft tissue tumors and solitary metastases and reported a mean MSTS in surviving patients of 72% (range, 33–87%) and a shoulder abduction ranging from 30°–45° (excluding one patient with no shoulder abduction at all) at 10-year-follow up [13].

While these functional results are more comparable to our reported results, the respective patient cohorts vary considerably. While Capanna et al. and Voggenreiter et al. included patients with soft tissue and secondary malignancies, our patient cohort solely consists of patients with primary malignant bone tumors [13,14]. In addition, patients reported by Capanna et al. and Voggenreiter et al. solely underwent proximal humeral replacement, whereas 3 of 6 surviving patients with available functional outcome in our cohort underwent total humeral replacement, making it again overall challenging to compare the functional outcome among these cohorts.

Although Grimer et al. showed that the proximal humerus had the lowest long-term rate of revision surgery compared to other locations in a cohort of 230 patients undergoing limb-salvage surgery with endoprosthetic replacement for a primary malignant bone tumor, the risk of endoprosthetic complications and revision surgery in musculoskeletal oncology is overall high with mechanical implant failures and periprosthetic infections being the predominant reasons for revision surgery [7,8]. Henderson et al. showed that although soft tissue failures (Henderson Type 1) only accounted for 12% of all failures in their cohort of 2174 patients receiving an endoprosthetic replacement for tumor resection, they accounted for 24% of endoprosthetic failures of proximal humerus replacements [7]. In addition, infection (Henderson Type 4) and tumor progression (Henderson Type 5) accounted for 37% and 17% of endoprosthetic failures of all proximal humerus replacements, respectively [7].

Although the relative distribution of failures described by Henderson et al. is in line with our cohort, the relative number of patients requiring revision surgery in our cohort (53%) is by far higher than reported by Henderson et al. (17%) [7]. This might be attributed to the additional demanding trapezius transfer in patients who might have otherwise undergone amputation.

Limb-survivorship following excision of bone tumors and subsequent megaprosthetic reconstruction is overall good to excellent as Jeys et al. reported a long-term amputation rate of only 8.9% (112 of 1261 patients) in these patients with local recurrence (63%; 71/112) and infection (34%; 38/112) being the “single biggest risk factor” for amputation [30,31]. Similar to Jeys et al., we observed 2 amputations (13%, 2/15), both attributed to a local recurrence. On the contrary, Capanna et al. observed only 1 patient (4%, 1/24) with a local recurrence in their cohort but opted against amputation in a palliative situation with concomitant inoperable lung metastases [14].

### Limitations

We acknowledge several limitations of our study. (1) The study follows a retrospective design and the reported endoprosthetic complications must be considered as low-end estimates as patients could have been treated elsewhere. (2) The number of patients in this study is small. While this can be attributed to the rarity of this approach, we are able—to the best of our knowledge—to report the largest cohort of patients undergoing a Bateman-type procedure in musculoskeletal oncology treated with a single design modular implant at a single tertiary university hospital. (3) Functional results were only available in 6 of 8 surviving patients as 1 patient lived overseas and 1 patient was not contactable. However, we repeatedly tried to contact the remaining patient but were unable to schedule a follow-up examination to determine shoulder ROM and determine the functional outcome scores. (4) Furthermore, there was no control group of patients that were treated with a different muscle transfer of free flaps for this indication.

## 5. Conclusions

The Bateman-type reconstruction is a feasible option for soft tissue reconstruction around megaprosthetic proximal or total humeral replacements in patients undergoing limb-sparing resection of a primary malignant bone tumor of the humerus. While the functional outcome is overall limited, good to excellent functional results can be achieved in individual patients. The risk for revision surgery is high within the first postoperative year but remains low thereafter. Limb-survivorship after the first and fifth postoperative year is overall acceptable.

## Figures and Tables

**Figure 1 cancers-13-03971-f001:**
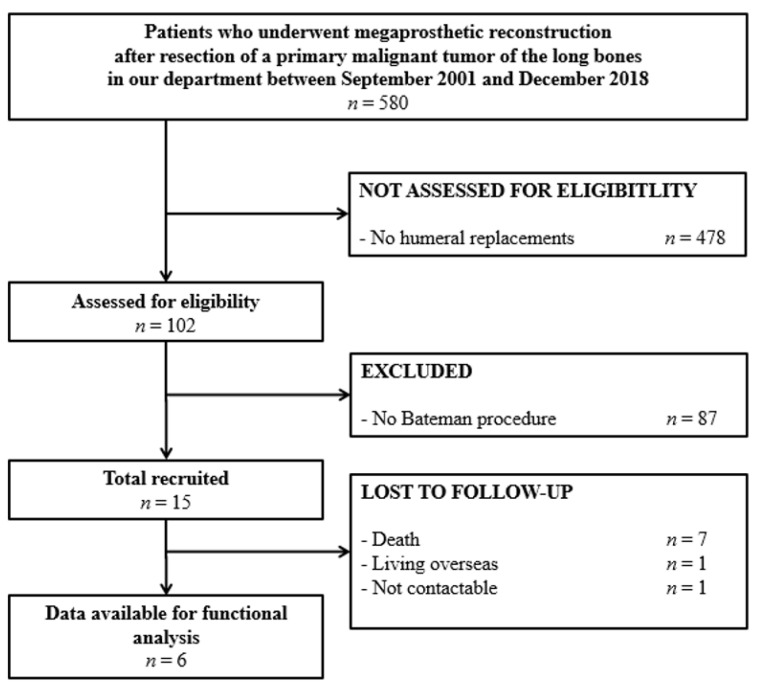
Study flow diagram.

**Figure 2 cancers-13-03971-f002:**
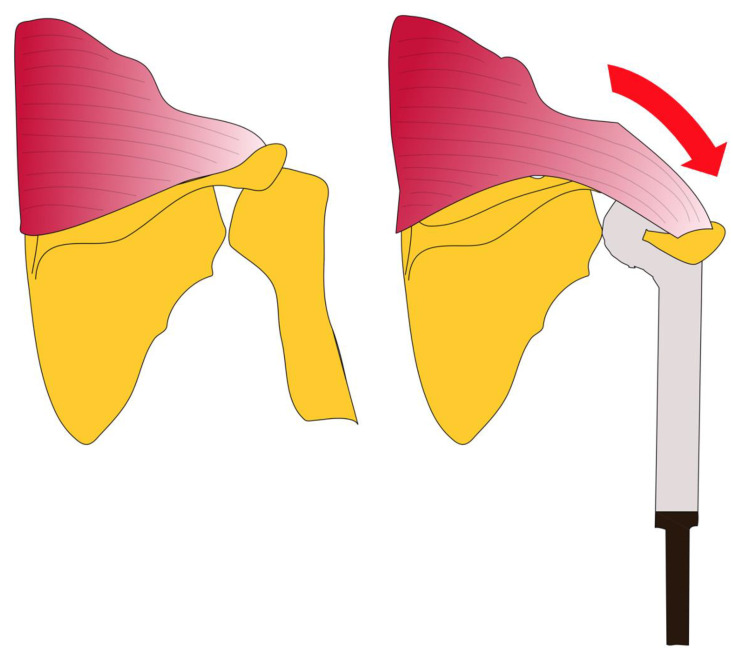
The Bateman-type reconstruction: The trapezius muscle and its acromial bony attachment are mobilized and attached to the attachment tube at the lateral proximal humerus.

**Figure 3 cancers-13-03971-f003:**
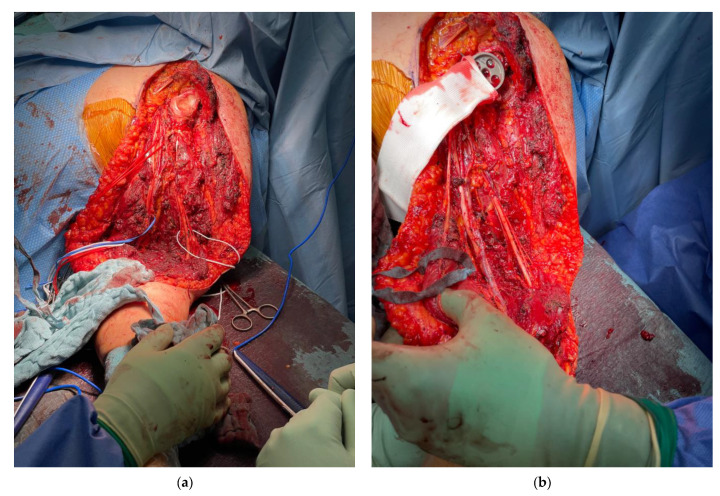
The Bateman-type soft tissue reconstruction with (**a**) the situs of the left upper extremity following glenohumeral dislocation and removal of the tumor with tagged neurovascular structures. (**b**) An attachment tube is fixed to the glenoid with non-absorbable No. 2 FibreWire^®^. (**c**) The assembled modular megaprosthesis is shuttled through the attachment tube and fixated with several FibreWire^®^ sutures before the acromial attachment of the trapezius muscle is identified and freed from deltoid muscle fibres. (**d**) The trapezius is then transferred with its acromial attachment and fixed to the attachment tube at the lateral proximal humerus using several FibreWire^®^ sutures.

**Table 1 cancers-13-03971-t001:** Exemplary overview of different surgical techniques to reconstruct soft tissues in proximal humeral endoprosthetic replacement.

Name	Surgical Technique
Bateman procedure [18]	Mobilization of the trapezius muscle and its bony attachments at the acromion and the lateral part of the scapular spine, fixing it to the lateral proximal humerus
Bateman-type reconstruction [17]	Mobilization of the trapezius muscle and its bony attachment at the acromion, fixing it to the attachment tube of the endoprothetic proximal humeral replacement
Gerber procedure [19]	Mobilization of the latissimus dorsi muscle and fixing it to the lateral proximal humerus
Tikhoff-Linberg procedure [13]	Extra-articular en bloc removal of the scapula, the lateral clavicle and the proximal humerus
Modified Tikhoff-Linberg procedure [14]	Resection of the glenoid, the lateral clavicle and the proximal humerus and endoprosthetic replacement of the proximal humerus
Sever-L’Episcopo procedure [11]	Mobilization of the latissimus dorsi and teres major muscles and transfer to the lateral proximal humerus posteriorly

**Table 2 cancers-13-03971-t002:** Demographics, oncological and surgical details.

Variable	*n* (%)
Gender	
Female	6 (40%)
Male	9 (60%)
Tumor Entity	
Osteosarcoma	10 (67%)
Chondrosarcoma	4 (27%)
Clear Cell Chondrosarcoma	1 (7%)
Type of Tumor Resection	
Intra-articular	6 (40%)
Extra-articular	9 (60%)
Type of Reconstruction	
Proximal Humeral Replacement	9 (60%)
Total Humeral Replacement	6 (40%)
Type of Shoulder Reconstruction	
Anatomic	12 (80%)
Reverse	3 (20%)

**Table 3 cancers-13-03971-t003:** Median MSTS, TESS and ASES scores differentiated by type of tumor resection and shoulder reconstruction.

	MSTS	TESS	ASES
Type of Tumor Resection			
Intra-articular	20	73	72
Extra-articular	20	66	73
	*p* = 0.7	*p* = 1.0	*p* = 0.7
Type of Shoulder Reconstruction			
Anatomic	18	52	65
Reverse	26	73	86
	*p* = 0.2	*p* = 0.1	*p* = 0.2

MSTS = Musculoskeletal Tumor Society Score, TESS = Toronto Extremity Salvage Score, ASES = American Shoulder and Elbow Surgeons Score.

**Table 4 cancers-13-03971-t004:** Shoulder ROM for patients undergoing anatomic and reverse shoulder reconstruction with statistical significance in bold.

	Ante- Version	Retro- Version	Abduction	Adduction	Internal Rotation	External Rotation
Type of Shoulder Reconstruction		10	10	20	30	15
Anatomic	10	10	10	20	30	15
Reverse	80	30	40	40	40	40
	***p*** **= 0.049**	***p*** **= 0.049**	***p*** **= 0.049**	***p*** **= 0.007**	*p***=** 0.112	***p*** **= 0.028**

ROM = range of motion.

## Data Availability

The data presented in this study are available on request from the corresponding author.

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
