# Peer review of "The Bateman-Type Soft Tissue Reconstruction around Proximal or Total Humeral Megaprostheses in Patients with Primary Malignant Bone Tumors—Functional Outcome and Endoprosthetic Complications"

_cancers, 2021, doi:10.3390/cancers13163971_

Round 1
Reviewer 1 Report
The manuscript on the Bateman-type reconstruction for proximal humeral megaprostheses is well written and of interest to musculoskeletal-tumour surgeons. My remarks are as follows: 1. Figures showing schematic drawings of the Bateman procedure seem more useful than the intraoperative pictures. 2. In their primary publication (reference no 16) the authors described a combination of the Bateman and the Gerber procedure using a latissimus dorsi flap. Has this combination been used in the present cases? In how many cases? Please describe and discuss this issue.Author Response
Dear Mr. Šekerović,
thank you for considering our paper for publication. Below we provide a point-by-point list of the reviewers’ comments and remarks, followed by our responses and changes to the manuscript (in italics, lines according to the revised manuscript “23072021RevisedManuscript.docx”).
Reviewer 1:
Figures showing schematic drawings of the Bateman procedure seem more useful than the intraoperative pictures.
- Thank you for pointing this out, we absolutely agree and have added an additional schematic drawing of the Bateman-type procedure to the manuscript (see Figure 2, Lines 172-174).
In their primary publication (reference no 16) the authors described a combination of the Bateman and the Gerber procedure using a latissimus dorsi flap. Has this combination been used in the present cases? In how many cases? Please describe and discuss this issue.
- As the experiences with a combining latissimus dorsi flap were non-satisfactory, we decided against performing an additional Gerber procedure but performed a sole Bateman-type reconstruction in all patients (Line 143-146).
The authors would like to thank the reviewers for their constructive comments and the time they invested in improving our manuscript.
Respectfully yours,
Kristian Schneider
Department of Orthopaedics and Tumor Orthopaedics
University Hospital of Münster, Germany

Reviewer 2 Report
The topic is interesting. The paper is well designed.
Even though the series is very small and heterogeneous, the paper merits publication after a few minor corrections.
A figure with description of the technique in Materials and Methods section would be an added value.
A recent review on Proximal humerus EPR should be added (Fiore M et al Arch Orthop Trauma Surg. 2021)
A table which include different techniques to reconstruct soft tissues in proximal humerus EPR should be added.
Author Response
Dear Mr. Šekerović,
thank you for considering our paper for publication. Below we provide a point-by-point list of the reviewers’ comments and remarks, followed by our responses and changes to the manuscript (in italics, lines according to the revised manuscript “23072021RevisedManuscript.docx”).
Reviewer 2:
A figure with description of the technique in Materials and Methods section would be an added value.
- We absolutely agree and have added an additional schematic drawing of the Bateman-type procedure (see Figure 2, Lines 172-174).
A recent review on Proximal humerus EPR should be added (Fiore M et al Arch Orthop Trauma Surg. 2021)
- Thank you for pointing this out, we have included this reference (see reference 6).
A table which includes different techniques to reconstruct soft tissues in proximal humerus EPR should be added.
- Thank you for pointing this out – we have added a table showing exemplary types of soft-tissue reconstruction in proximal humerus EPR (see Table 1, Lines 60-62).
The authors would like to thank the reviewers for their constructive comments and the time they invested in improving our manuscript.
Respectfully yours,
Kristian Schneider
Department of Orthopaedics and Tumor Orthopaedics
University Hospital of Münster, Germany
Reviewer 3 Report
This well written paper describes a rarely used method of reconstruction following proximal humeral replacement. The results (very poor) speak for themselves - and explain why it is not used more!
Only one query - how were the patients who underwent this procedure selected - did they have particular risk factors that may explain the high complication rate - eg. complete resection of deltoid?
Author Response
Dear Mr. Šekerović,
thank you for considering our paper for publication. Below we provide a point-by-point list of the reviewers’ comments and remarks, followed by our responses and changes to the manuscript (in italics, lines according to the revised manuscript “23072021RevisedManuscript.docx”).
Reviewer 3:
Only one query - how were the patients who underwent this procedure selected - did they have particular risk factors that may explain the high complication rate - eg. complete resection of deltoid?
- Patients were selected for the Bateman-type reconstruction based on soft-tissue coverage following tumor resection with wide surgical margins. So, the reviewer is absolutely right when he points out that these specific risk factors lead to the high complication rate. We have specified this within the manuscript (Line 139-143; Line 283-286).
The authors would like to thank the reviewers for their constructive comments and the time they invested in improving our manuscript.
Respectfully yours,
Kristian Schneider
Department of Orthopaedics and Tumor Orthopaedics
University Hospital of Münster, Germany
Round 2
Reviewer 1 Report
Thank you for improvements. Nice manuscript.